# A Plant-Derived Maternal Vaccine against Porcine Epidemic Diarrhea Protects Piglets through Maternally Derived Immunity

**DOI:** 10.3390/vaccines11050965

**Published:** 2023-05-09

**Authors:** Eun-Ju Sohn, Hyangju Kang, Kyungmin Min, Minhee Park, Ju-Hun Kim, Hwi-Won Seo, Sang-Joon Lee, Heeyeon Kim, Dongseob Tark, Ho-Seong Cho, Bo-Hwa Choi, Yeonsu Oh

**Affiliations:** 1BioApplications Inc., Pohang Techno Park Complex, 394 Jigok-ro, Pohang 37668, Republic of Korea; ejsohn@bioapp.co.kr (E.-J.S.); hyangju@bioapp.co.kr (H.K.); mkm8825@bioapp.co.kr (K.M.); pmhee@bioapp.co.kr (M.P.); khjh99@bioapp.co.kr (J.-H.K.); bhchoi@bioapp.co.kr (B.-H.C.); 2Infectious Disease Research Center, Korea Research Institute of Bioscience and Biotechnology, Daejeon 34141, Republic of Korea; seohw01@kribb.re.kr; 3College of Veterinary Medicine and Institute of Veterinary Science, Kangwon National University, Chuncheon 24341, Republic of Korea; sjoon516@kangwon.ac.kr (S.-J.L.); hykim9269@korea.kr (H.K.); 4Korea Zoonosis Research Institute, Jeonbuk National University, Iksan 54531, Republic of Korea; tarkds@jbnu.ac.kr; 5College of Veterinary Medicine and Bio-Safety Research Institute, Jeonbuk National University, Iksan 54596, Republic of Korea; hscho@jbnu.ac.kr

**Keywords:** plant-derived vaccine, PEDV, sow vaccine

## Abstract

Newborn piglets are susceptible to a highly contagious enteritis caused by the porcine epidemic diarrhea virus (PEDV), associated with high levels of mortality worldwide. There is pressing need for a rapid, safe, and cost-effective vaccine to safeguard pigs from getting infected by PEDV. PEDV belongs to the coronavirus family and is characterized by high levels of mutability. The primary goal of a PEDV vaccine is to provide immunity to newborn piglets through vaccination of sows. Plant-based vaccines are becoming more popular because they have low manufacturing costs, are easily scalable, have high thermostability, and a long shelf life. This is in contrast to conventional vaccines which include inactivated, live, and/or recombinant types that can be expensive and have limited ability to respond to rapidly mutating viruses. The binding of the virus to host cell receptors is primarily facilitated by the N-terminal subunit of the viral spike protein (S1), which also contains several epitopes that are recognized by virus-neutralizing antibodies. As a result, we generated a recombinant S1 protein using a plant-based vaccine platform. We found that the recombinant protein was highly glycosylated, comparable to the native viral antigen. Vaccination of pregnant sows at four and two weeks before farrowing led to the development of humoral immunity specific to S1 in the suckling piglets. In addition, we noted significant viral neutralization titers in both vaccinated sows and piglets. When challenged with PEDV, piglets born from vaccinated sows displayed less severe clinical symptoms and significantly lower mortality rates compared to piglets born from non-vaccinated sows.

## 1. Introduction

Neonatal piglets are highly susceptible to porcine epidemic diarrhea virus (PEDV), a member of the *Alphacoronavirus* genus within the *Coronaviridae* family of the *Nidovirales* order. PEDV infection is characterized by acute diarrhea, vomiting, dehydration, and often leads to high mortality rates [1]. For the last thirty years, PEDV has been reported in the pig industries of Europe and Asia. The initial case was observed in England [2], and the first isolation of the virus in Belgium [3]. Subsequently, Asian countries with pig farming have now become endemic to PEDV. In the United States, the first report of PEDV was in 2013 and within months it had quickly spread across the country, causing significant economic damage [4,5]. Infected piglets exhibit watery diarrhea within the first few weeks of life, accompanied by dehydration, vomiting, and anorexia, which result in high morbidity and mortality rates [6]. This is the characteristic presentation of PEDV.

The genome of PEDV comprises five open reading frames (ORFs), which encode four structural proteins (the spike protein S, envelope protein, membrane protein, and nucleocapsid protein) and three non-structural proteins (the replicases ORF1a and 1b, and ORF3) [7]. Located on the surface of the PEDV virion, the S protein plays a crucial role in attaching the virus to host cell receptors [8,9]. Furthermore, due to the presence of viral neutralizing epitopes, the S protein serves as the principal antigenic determinant and is a target for neutralizing antibody responses. The S protein naturally forms a homotrimeric structure and contains several predicted sites for glycosylation [10].

Lactogenic immunity is the primary mechanism of antiviral protection in neonatal piglets. During lactation, colostrum and milk serve as a medium for passive transfer of sIgA, IgG, and IgM to the piglet [11]. Despite attempts to create a vaccine against PEDV, there is currently no effective vaccine available on the market to safeguard newborn piglets [1]. Immunizing pregnant sows is crucial for controlling the spread of PEDV and reducing the mortality rate of suckling piglets [12]. Therefore, the primary goal of the PEDV vaccine development is to provide immunity to newborn piglets through vaccination of sows. Conventional inactivated, live, and/or recombinant vaccines are costly and are limited in their ability to respond to a quickly mutating virus [13]. Plant-based vaccines are attracting increasing attention due to their low manufacturing cost, ease of scaling up production, high thermostability, and long shelf life [14].

In this study, we synthesized a codon-optimized, full-length gene for the PEDV S1 protein fused to a porcine Fc (pFc2) domain. The Fc domains of antibodies promote effector function, whereas the Fab domains are responsible for targeting. Fc engineering can modify effector functions, such as antibody dependent cellular cytotoxicity (ADCC), antibody dependent cellular phagocytosis (ADCP), and serum half-life [15]. The vaccine developed in this study was evaluated for safety, efficacy, and potency in an experimental facility.

## 2. Materials and Methods

### 2.1. Generating Plasmid

For expression of NBH1:PEDV S1:pFc2 in *Nicotiana benthamiana*, coding sequences of the PEDV S1 protein (Genbank AHZ45711.1; amino acids 21-736) and porcine Fc domain (pFc2, Genbank BAM66310.1; amino acids 97-323) were synthesized after codon-optimization for expression in *N. benthamiana*. There is a 99.7% similarity in the amino acid sequence between the S1 domain of the vaccine strain (AHZ45711.1) and the challenge strain (AOS52336.1). To generate the construct NBH1:PEDV S1:pFc2, PEDV S1, and pFc2, DNA fragments were inserted downstream of the 6X histidine tag in the pCAMBIA1300 binary vector containing a cauliflower mosaic virus (CaMV) 35S promoter, a BiP signal sequence fused with 6X histidine tag, and a heat shock protein (HSP) terminator following restriction digestion with *BamH*I, *Xma*I and *Xma*I, *Sac*I, respectively [16].

### 2.2. Production of NBH1:PEDV S1:pFc2

To express NBH1:PEDV S1:pFc2 in *N. benthamiana*, the binary vector NBH1:PEDV S1:pFc2 was transformed into *Agrobacterium tumefaciens* strain LBA4404 before vacuum infiltration of 5-week-old *N. benthamiana* plants [16,17]. To enhance the expression level of NBH1:PEDV S1:pFc2, the gene silencing suppressor p38 was co-infiltrated [17]. Infiltrated leaves were harvested after 4 days, frozen in liquid nitrogen, and ground into a powder using a pestle and mortar. For purification, powders were resuspended in extraction buffer (50 mM Sodium phosphate pH 8.0, 300 mM NaCl, 50 mM Glycine, 0.5 % Triton X-100) in a 1:2 ratio (*w/v*), before debris was removed by centrifugation at 17,000× *g* at 4 °C for 40 min. Total protein extracts were loaded onto a column filled with 0.1× volume of protein A agarose (Amicogen, Jinju, Republic of Korea). The protein A agarose was subsequently washed twice with 0.5× volume of wash buffer (50 mM Sodium phosphate pH 8.0, 300 mM NaCl). The recombinant protein was then eluted using 0.5× volume of elution buffer (100 mM sodium citrate pH 3.0, 300 mM NaCl), and the pH was adjusted to 7.4 using 1.5 mM Tris-Cl pH 8.8. Protein samples from each purification fraction were subject to western blot analysis using horseradish peroxidase (HRP) conjugated antipig IgG antibody (Bethyl Laboratories, Montgomery, TX, USA). Purified NBH1:PEDV S1:pFc2 was analyzed by sodium dodecyl sulfate-polyacrylamide gel electrophoresis (SDS-PAGE).

### 2.3. Size-Exclusion Chromatography

Size-exclusion chromatography assays were performed using the NGC chromatography system (Bio-Rad, Hercules, CA, USA) and a Superose 6 increase 10/300 gl column (GE Healthcare, Chicago, IL, USA). To prepare for the loading of PEDV S1 protein, the column was washed and equilibrated using equilibration buffer (EB; 25 mM Tris-Cl pH 8.0, 100 mM NaCl) at a flow rate of 0.75 mL/min. Absorbance at 280 nm was monitored throughout the process, and fractions were collected and then subjected to 8 % SDS-PAGE. Thyroglobulin (660 kDa), Ferritin (440 kDa), Aldolase (158 kDa), and Ovalbumin (44 kDa) (GE Healthcare, Chicago, IL, USA) were used as size standards.

### 2.4. Endoglycosidase H Treatment Assay

Purified NBH1:PEDV S1:pFc2 was treated with Endo H (New England Biolabs, Beverly, MA, USA) according to the manufacturer’s instructions. Briefly, 2.5 μg of NBH1:PEDV S1:pFc2 was mixed with 1 μL of 10X glycoprotein denaturing buffer (5% SDS, 400 mM DTT) and distilled water to make a 10 μL reaction volume. The mixture was then boiled for 10 min to facilitate protein denaturation, followed by cooling on ice. Next, 2 μL of 10X glycobuffer 3 (500 mM sodium acetate, pH 6), 1 μL of Endo-H, and 7 μL of distilled water were added to the reaction mixture, which was then incubated at 37 °C for 1 h. Samples were analyzed by SDS-PAGE.

### 2.5. Vaccine Preparation

The final test vaccine was prepared with 500 μg of antigen with Montanide ISA 15A VG after testing 500 or 800 μg of antigen and various adjuvants, including 1 mg/mL Carbopol^®^ 971P NF Polymer(Lubrizol, Cleveland, OH, USA), 20% (*v/v*) Emulsigen-D (MVP Adjuvant, Omaha, NE, USA), 15% (*v/v*) Montanide ISA 15A VG (Seppic, La Garenne-Colombes, France), 50% (*v/v*) CASq (Huvet Bio, Seoul, Republic of Korea), and 50% (*v/v*) Montanide IMS1313 VG N (Seppic, La Garenne-Colombes, France), using guinea pigs.

### 2.6. Experimental Design

Twelve pregnant sows of conventional crossbreed were purchased from a PEDV-free pig farm. Prior to the study, the sows tested negative for both serology and antigenicity against PEDV. The sows were then transferred to a research facility, where they were housed individually in separate rooms. The allocation of sows into groups, either vaccinated or non-vaccinated, was done randomly. Vaccinated sows (V-SOW, *n* = 6) were administered the vaccine intramuscularly at 6 and 2 weeks antepartum. Vaccinated sows were observed for clinical symptoms including anorexia, diarrhea, abortion, suppuration, and necrosis at the inoculation site after every vaccination. Non-vaccinated sows (NV-SOW, *n* = 6) served as controls. At parturition, sows farrowed naturally, and no cross-fostering was permitted to ensure maternal antibody transfer into the blood of piglets. The six vaccinated sows delivered 62 live piglets, while the six non-vaccinated sows delivered 64 live piglets. A total of 48 newborn piglets (four piglets per sow) were randomly selected on the day of birth and were monitored to ensure they were able to suckle colostrum and milk. Among the remaining piglets, ten served as a negative control without challenge.

Five days after birth, all piglets were challenged orally with 10^4.0^ TCID_50_/_mL_ PEDV genotype G2b (QIAP 1401-p70; AOS52336.1) [18]. The animals were regularly sampled, and clinical signs including mortality, body temperature, fecal condition, and body weight were monitored until the end of the experiment. The clinical score was determined based on diarrhea severity: 0, no clinical signs (healthy), 1, moderate diarrhea, and 2, severe diarrhea, vomiting, and dehydration [18]. All piglets were anesthetized at 15 dpc, and jejunum tissue samples were collected at necropsy and were subject to histopathological and immunohistochemistry analysis.

### 2.7. Sample Collection

Blood samples from sows were collected before the first and second vaccination, and at farrowing. At farrowing, colostrum of each sow was also collected. Fecal samples were collected from piglets for 5 consecutive days from the day of challenge and then again on the 7th, 9th, 12th, and 15th day post-challenge.

### 2.8. Fecal Viral Antigen Detection

Piglet fecal swabs were treated using QIAamp Viral RNA Kit (Qiagen, Hilden, Germany), and samples were subsequently subject to real-time quantitative PCR (SensiFAST Probe No-ROX One-Step Kit, Meridian Bioscience, OH, USA) according to the manufacturers’ instructions. The primers and probe for real time PCR used to identify PEDV antigens are as follows: F, CGCAAAGACTGAACCCACTAATTT; R, TTGCCTCTGTTGTTACTTGGAGAT; and probe, FAM-TGTTGCCATTGCCACGACTCCTGC-BHQ1.

### 2.9. Serological Analysis: Virus Neutralizing (VN) Antibody and Antibody ELISA Titers

VN titers in sow serum and colostrum were determined as previously described, with the following modifications [6,18]. Briefly, for colostrum samples, 1 mL of colostrum was incubated with 30 µL rennet (5 mg/mL, Sigma-Aldrich, St. Louis, MO, USA) at 37 °C for 1 h. The solidified whey was separated by centrifugation at 6000× *g* for 20 min. Whey samples were four-fold diluted in PBS containing 0.05% Tween 20 and 1% casein, while serum samples were diluted two-fold (ranging from 1:4 to 1:512). The diluted serum and colostrum samples were subsequently mixed with an equal volume of a virus suspension containing 100–200 TCID_50_/100 µL PEDV (QIAP 1401-p70). After 1 h of incubation at 37 °C, 200 µL of the virus–serum mixture was added to confluent Vero cell monolayers, incubated for 1 h at 37 °C in 5% CO_2_, washed three times with PBS containing 1 µg/mL trypsin, and replenished with viral growth medium. After another 48 h incubation, VN titers were calculated as reciprocal values of the highest serum/colostrum dilution that inhibited PEDV-specific CPE.

For the antibody ELISA titers in sow serum, ELISA plates were coated overnight at 4 °C with 100 µL/well of inactivated whole virus (10^2.5^ TCID_50_/_mL_) in PBS. To perform the ELISA assay, the plates were first washed five times with PBS containing 0.05% Tween 20 to remove any unbound material. The pig sera were then serially diluted in an assay diluent buffer (PBS with 0.05% Tween 20, 1% fish gelatin) and added to coated wells. Following a 2-h incubation, the plates were washed again and incubated with HRP-conjugated antipig IgG antibody (diluted 1/3000) for 1 h. The plates were then washed again and developed with a 1 Step Ultra TMB ELISA substrate solution. The reaction was stopped by adding 30 µL of 2 M sulphuric acid to each well, and the absorbance at 450 nm was measured using a microtiter plate reader.

### 2.10. Histopathological Analysis

For morphometric analysis of histopathological changes in the intestine, three sections were examined blindly. The lesion score was determined as previously described with some modifications [1,19], as summarized in Table 1.

### 2.11. Immunohistochemistry (IHC) Analysis

IHC was performed as previously described [20] using an anti-PEDV-S1 monoclonal antibody (Creative Diagnostics, Shirley, NY, USA). Ten randomly selected fields of individual formalin-fixed jejunum sections were subject to morphometric analysis. The signal score was assigned as follows: 0, none or <10% of field area; 1, 10–30% of field area; 2, 30–50% of field area; and 3, >50% of field area.

### 2.12. Statistical Analysis

Summary statistics were calculated for all groups to assess the overall quality of the data, including normality. The statistical relationship of clinical sign scores, viral RNA quantification, serological results, and histopathological results between the vaccinated and control groups were determined with a student’s t-test, followed by the Mann–Whitney U test and Wilcoxon rank-sum test to calculate the post-power of the findings using IBM SPSS Statistics (ver. 24, New York, NY, USA). Differences were considered significant if *p* < 0.05.

## 3. Results

### 3.1. Expression and Purification of NBH1:PEDV S1:pFc2 from N. benthamiana

PEDV S1 and porcine Fc domain gene sequences were codon-optimized for expression in *N. benthamiana*. The porcine Fc region was used to: (1) enable the purification of ppS1-Fc by affinity chromatography using protein A agarose and (2) increase the stability of S1 in animals [21]. The S1-Fc fusion protein was targeted to the ER by using the BiP signal sequence, and the ER retention signal HDEL was used to increase accumulation. A 6-histidine tag was fused between the BiP signal peptide and S1 for high expression (data not displayed). The chimeric gene was expressed under control of the CaMV 35S promoter and HSP terminator to enhance protein expression (Figure 1A). The expression vector was transformed into *Agrobacterium* and subsequently vacuum infiltrated into *N. benthamiana*. Proteins were extracted from the infiltrated leaves at 4 dpi and subject to fractionation and western blot analysis using an HRP-conjugated antipig IgG antibody. Almost all of the NBH1:PEDV S1:pFc2 protein was expressed in the soluble fraction with a size of 130 kDa (Figure 1B). Although NBH1:PEDV S1:pFc2 was predicted to be 105 kDa from the amino acid sequence, it ran at 130 kDa due to the presence of 13 putative N-glycosylation sites (12 in PEDV S1 domain and 1 in pFc2). Consistent with the predicted size of NBH1:PEDV S1:pFc2, S1 proteins previously expressed in insect and mammalian cells were 100 and 130 kDa, respectively [6]. Additionally, PNGase F treatment was found to decrease the size of proteins to 80 kDa, indicating that the difference in size observed following western blotting of S1 between organisms is due to the differences in N-glycosylation [6].

Leaves were harvested at 4 dpi and subject to protein purification. Protein was sampled at all steps of purification, and the efficiency of purification was analyzed by western blotting using the HRP-conjugated antipig IgG antibody. The NBH1:PEDV S1:pFc2 protein bound to and could be eluted from protein A agarose. Only a small amount of protein was present in the flow-through (FT) fraction and on beads after elution (EB) (Figure 1C). The purity of ppS1-Fc was examined by SDS-PAGE, followed by Coomassie Brilliant Blue (CBB) staining. The eluted fraction contained proteins of 130 and 70 kDa (Figure 1D). Most of the protein ran at 130 kDa, which was referred to as NBH1:PEDV S1:pFc2. The 70 kDa protein was less abundant and seemed to be a truncated form of NBH1:PEDV S1:pFc2, as this band was also detected by western blot analysis (Figure 1C). Overall, 88 mg of NBH1:PEDV S1:pFc2 was purified from 1 kg *N. benthamiana leaves* (fresh weight) transiently transformed with the *NBH1-PEDV S1:pFc2* construct.

### 3.2. Trimeric Structure Characterization and Glycosylation Analysis

To investigate whether NBH1:PEDV S1:pFc2 formed a trimer, purified protein was analyzed by size exclusion chromatography and each fraction was also analyzed by SDS-PAGE, followed by Coomassie blue staining. The main peak observed in the NBH1:PEDV S1:pFc2 chromatogram was a similar size to 440 kDa ferritin, indicating that the majority of NBH1:PEDV S1:pFc2 were arranged as trimers, since the expected size of an NBH1:PEDV S1:pFc2 trimer was 390 kDa (Figure 2A,B).

NBH1:PEDV S1:pFc2 was designed for accumulation in the endoplasmic reticulum (ER), indicating that the protein would be N-glycosylated. To confirm whether NBH1:PEDV S1:pFc2 was glycosylated and accumulated in the ER, we treated samples with endoglycosidase H (Endo H) and subsequently analyzed them by SDS-PAGE. Almost all NBH1:PEDV S1:pFc2 was sensitive to Endo H, indicating that this protein accumulated in the ER and was N-glycosylated (Figure 2C).

### 3.3. Sow Immunogenicity: VNT in Serum and Colostrum, Antibody ELISA Titer

There was a significant (*p* < 0.05) difference in the log-transformed VNT titers of serum and colostrum antibodies between vaccinated and non-vaccinated sows (Figure 3A,B). We also observed a significant (*p* < 0.05) difference in serum antibody ELISA titers probing PEDV between vaccinated and non-vaccinated sows (Figure 3C).

### 3.4. Fecal Viral Shedding Detection Rate and Mortality

Fecal viral RNA was not detected in either vaccinated or non-vaccinated sows at 0 dpc. The PEDV positivity rate in piglets from vaccinated sows was 63% at 1 dpc and 100% at 3 dpc, lasting for 2 days before the positivity rate began to decrease from 4 dpc. No virus shedding was detected at 12 dpc and 15 dpc. Piglets from non-vaccinated sows displayed a positivity rate of 71% at 1 dpc and 100% at 2 to 5 dpc. The positive rate subsequently decreased but remained as high as 17% at 15 dpc.

The mortality rate was significantly reduced to 20% in piglets from vaccinated sows compared to 50% in piglets from non-vaccinated sows at 15 dpc (*p* < 0.05; Table 2).

### 3.5. Histopathological and Immunohistochemical Analyses

The height of the intestinal villi (VH), which is directly related to nutrient absorption, the depth of the crypt (CD), which is related to the continuous regeneration of enterocytes, and the ratio of both lengths (VH:CD) were significantly higher in piglets of vaccinated sows (*p* < 0.05) (Table 3).

The results of the study demonstrated that the vaccinated piglets had significantly lower histopathological lesion and antigen scores compared to the piglets of non-vaccinated sows (*p* < 0.05) (Figure 4).

## 4. Discussion

PEDV infections are a significant cause of morbidity in piglets, with mortality rates approaching 100% [19]. Despite the current use of inactivated and live attenuated vaccines, these measures have had limited success. The recommended immunization schedule involves two doses of attenuated live or inactivated vaccine in gilts at 2–3 week intervals before service and in pregnant sows between 12 and 14 weeks of gestation [22]. However, the high frequency of recombination events observed in coronaviruses may result in the generation of genetically diverse viruses, which may alter virulence during infections [23]. The emergence of highly virulent strains and recurrent outbreaks, which are observed even in vaccinated farms, highlight the need for more effective vaccines. This means that vaccines with high protective antigens should continuously be monitored with the current antigenic situation, responding to mutations over time.

The plant vaccine platform has advantages in the context of PEDV, including use as a rapid response to viral antigenic change. We have demonstrated enhanced immunogenicity of a PEDV antigen using molecular engineering. The Fc immunoglobulin domain was employed as an antigen delivery platform with enhanced avidity of binding to an Fc receptor. This is an improvement on conventional IgG, which binds the Fc receptor to opsonize antigens, and IgM, which despite being polymeric, displays low binding affinity for its receptor [24]. The most antigenic region of PEDV is the S protein, which is a target of neutralizing antibodies in the host and facilitates viral binding to host cell receptors [6,25]. In this study, the full-length, codon-optimized S1 gene was stably expressed and highly glycosylated in our plant vaccine production platform. Protein glycosylation is a critical and intricate post-translational modification that plays essential roles in biological activity, including solubility, stability, cellular localization, molecular trafficking, self-recognition, clearance, transport, immunogenicity, and circulating half-life. Such abundant glycosylation is an advantage of the plant-based vaccine production system that cannot be achieved in bacterial expression systems [26,27,28].

We evaluated the safety and protective efficacy of our recombinant vaccine against PEDV in newborn piglets. In immunized pregnant sows, we observed elicited antibody ELISA and viral neutralizing titers in the serum and high levels of neutralizing antibodies in the colostrum collected after farrowing. It is promising to see that the plant-derived S1 with Fc domain subunit vaccine provided effective lactogenic immunity, as demonstrated by the survival of neonatal piglets from immunized sows after challenge with virulent PEDV. This suggests that the vaccine is capable of inducing protective antibodies that were transferred to the piglets through colostrum and milk, providing them with the necessary immunity to survive the viral challenge. However, further studies are needed to fully assess the safety, efficacy, and long-term immunity conferred by this vaccine. In addition, histopathological examination at the experimental end point demonstrated that lesion scores of piglets from vaccinated sows were significantly lower than unvaccinated animals. Further, the duration of virus shedding into the feces was significantly shorter in the piglets of vaccinated sows. These results suggest that vaccination reduces clinical signs of disease in piglets, and may reduce the risk of transmission between groups by blocking viral shedding. The results of this study demonstrate the utility of our highly antigenic, recombinant vaccine composed of protein combined with a molecular adjuvant. In particular, veterinary vaccine manufacture may benefit from this technology due to less stringent GMP requirements and the associated costs of production, and the attractive possibility of vaccinating orally, particularly in the case of gastrointestinal pathogens [29]. Further studies are required with a larger number of pigs in the field, and to optimize immunization procedures. The recombinant S1 protein with Fc domain has potential for effectively preventing PEDV.

## Figures and Tables

**Figure 1 vaccines-11-00965-f001:**
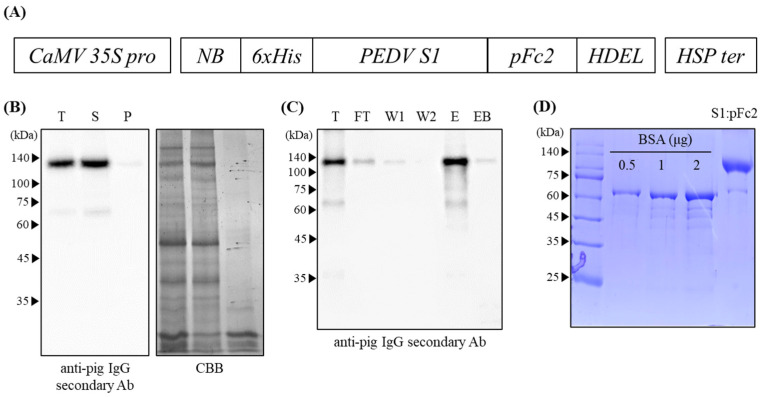
Schematic representation of the *PEDV S1:pFc2* construct and production of *PEDV S1:pFc2* from *N. benthamiana* leaves. (**A**) Schematic representation of the *NBH1:PEDV S1:pFc2* construct. CaMV 35S pro, cauliflower mosaic virus *35S* promoter; NBH1, signal peptide from Binding Protein and 6 His-tag; PEDV S1, porcine epidemic diarrhea virus S1 protein; pFc2, porcine Fc domain; HDEL, ER retention signal; and HSP ter, heat shock protein terminator. (**B**) Western blot analysis after sup-pellet fractionation. T, total; S, supernatant; and P, pellet. (**C**) Western blot analysis of protein fractions obtained at different steps of the purification process. T, total; FT, flow-through; W1, wash 1; W2, wash 2; E, eluate; and EB, protein sample eluted from protein A agarose. (**D**) SDS-PAGE analysis of purified NBH1:PEDV S1:pFc2. Bovine serum albumin (BSA) was used as a loading control to determine the amount of purified NBH1:PEDV S1:pFc2. kDa, kilo Dalton.

**Figure 2 vaccines-11-00965-f002:**
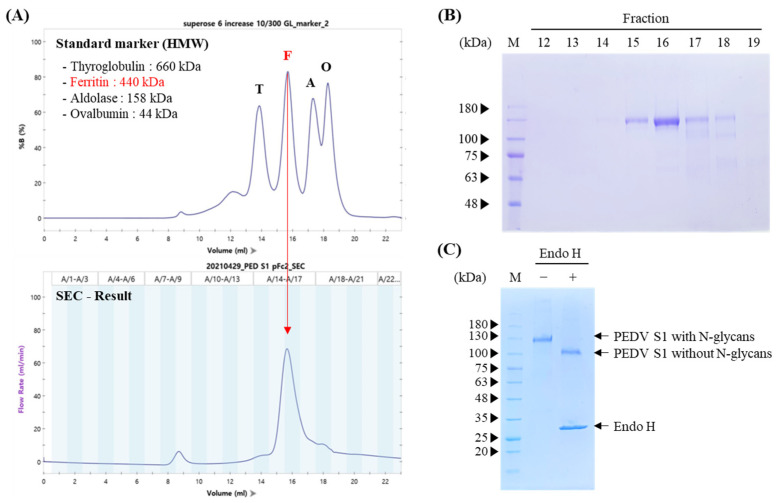
Size exclusion chromatography and N-glycosylation of NBH1:PEDV S1:pFc2. (**A**) Chromatogram of NBH1:PEDV S1:pFc2 and high-molecular weight standard marker. T, thyroglobulin; F, ferritin; A, aldolase; and O, ovalbumin. (**B**) SDS-PAGE analysis of each fraction form size exclusion chromatography of NBH1:PEDV S1:pFc2. (**C**) Analysis of Endo H treated purified NBH1:PEDV S1:pFc2 by SDS-PAGE and Coomassie blue staining.

**Figure 3 vaccines-11-00965-f003:**
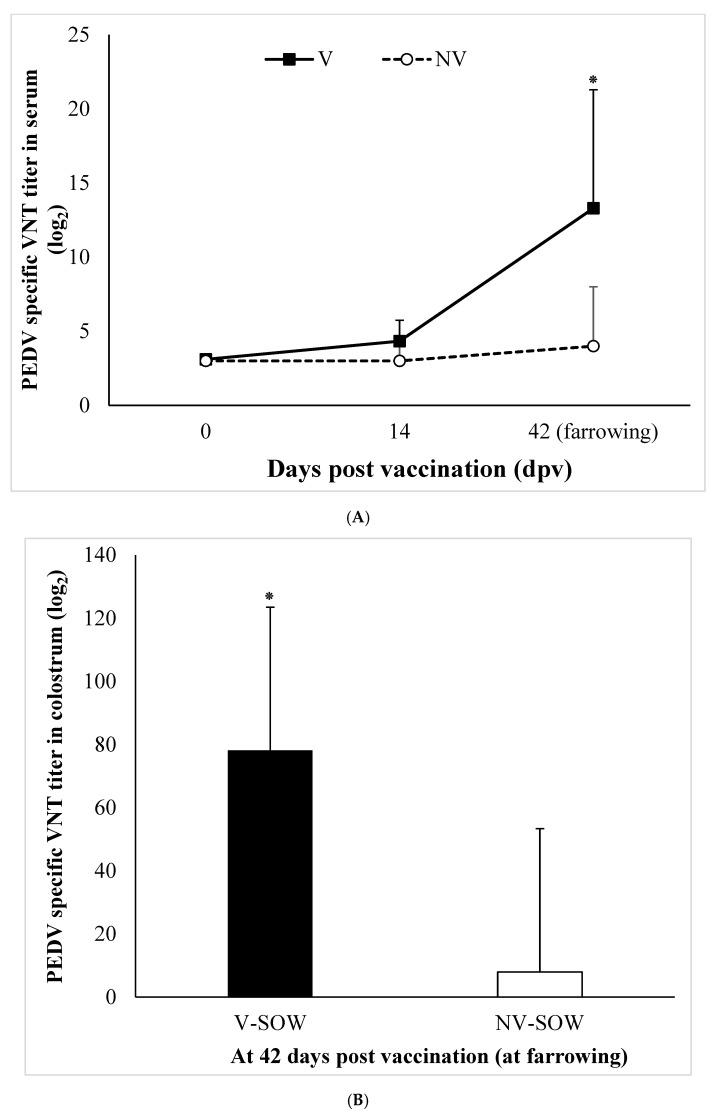
PEDV specific neutralizing antibody titers in serum (**A**) and colostrum (**B**) samples of sows, and PEDV antibody ELISA titers in serum of sows at different time points (**C**). Bars represent mean ± standard deviation (SD) during animal experiment. *, *p* < 0.05.

**Figure 4 vaccines-11-00965-f004:**
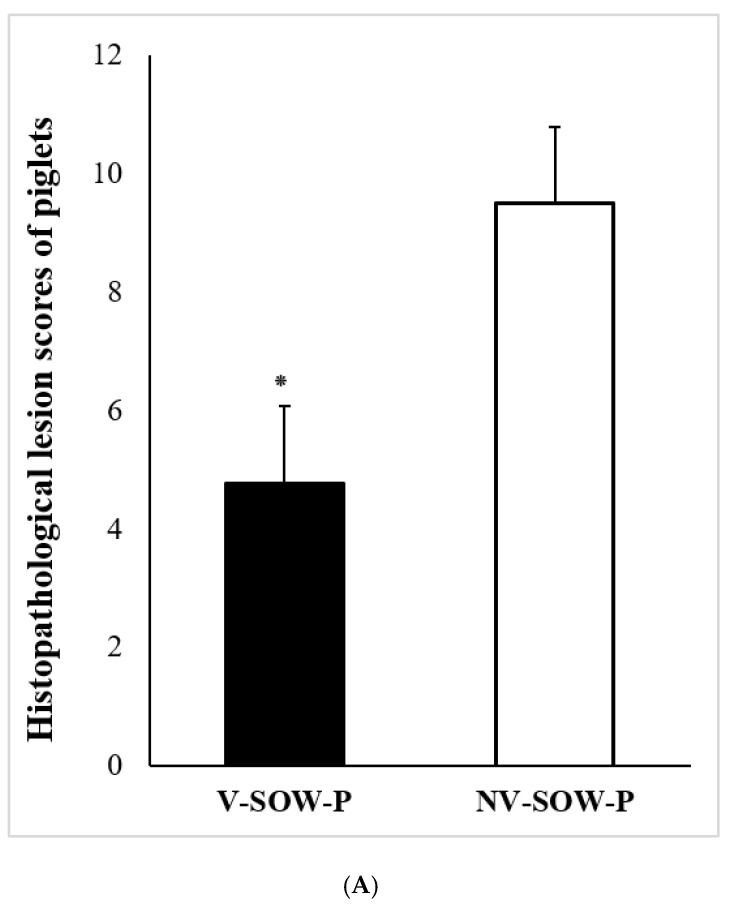
Comparative analysis of histopathological lesion scores (**A**) and immunohistochemical antigen scores (**B**) between groups, V-SOW-P and NV-SOW-P. 90 sites per group were analyzed. *, *p* < 0.05.

**Table 1 vaccines-11-00965-t001:** Histopathological lesion criteria and scoring.

1. Histopathological Classification	2. Scoring System				
	0	1	2	3	4
Villous atrophy	Normal, villus height to crypt depth ratio at least 3:1	Mild villus atrophy; villous height to crypt ratio less than 3:1	Moderate villus atrophy; villous to crypt ratio 1:1 to 2:1	Severe villus atrophy; villous height to crypt depth ratio less than 1:1 without villi or crypt damage	Severe villus atrophy; villous height to crypt depth ratio less than 1:1 with villi and crypt loss
Villous epithelial vacuolation	Absent	Vacuolation in <25% of villus epithelium	Vacuolation in 25% to 50% of villus epithelium	Vacuolation in >50% of villus epithelium	Necrotic villus epithelium with or without crypt loss
Necrotic cells	None to minimal necrotic cells (<10%)	Minimal necrotic cells (<25%)	Moderate necrotic cells (25% to 50%)	Marked necrotic cells (>50%)	
Hyperemia in lamina propria	None to minimal hyperemia (<10%)	Minimal hyperemia (<25%)	Moderate hyperemia (25% to 50%)	Marked hyperemia (>50%)	

**Table 2 vaccines-11-00965-t002:** Fecal virus shedding and mortality.

	Days Post Challenge									
	0	1	2	3	4	5	7	9	12	15
Viral shedding										
V-SOW-P	0/24 ^†^	15/24	23/23	22/22	18/20	9/19	7/19	3/19	0/19	0/19
Positive rate	0%	63%	100%	100%	90%	47% *	37% *	16% *	0% *	0% *
NV-SOW-P	0/24	17/24	21/23	18/18	18/18	14/14	10/13	8/12	8/12	2/12
Positive rate	0%	71%	100%	100%	100%	100%	77%	67%	67%	17%
Mortality										
V-SOW-P	0	0	1	1	2	1	0	0	0	0
	Total mortality rate: 19/24 (20%) *									
NV-SOW-P	0	0	1	5	0	4	1	1	0	0
	Total mortality rate: 12/24 (50%)									

*, *p* < 0.05; ^†^, positive number/total number.

**Table 3 vaccines-11-00965-t003:** Mean villous height (VH) and crypt depth (CD) between groups.

Group	Mean VH (µm) ± SD	Mean CD (µm) ± SD	VH:CD, Mean ± SD
**V-SOW-P**	116.54 ± 44.05 *	153.28 ± 27.62 *	0.81 ± 0.38 *
**NV-SOW-P**	71.56 ± 38.12	140.25 ± 51.18	0.50 ± 0.23

Sixty sites per group were analyzed. *, *p* < 0.05.

## Data Availability

Data sharing is not applicable to this article.

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
