# Peer review of "A Plant-Derived Maternal Vaccine against Porcine Epidemic Diarrhea Protects Piglets through Maternally Derived Immunity"

_vaccines, 2023, doi:10.3390/vaccines11050965_

Round 1

Reviewer 1 Report

The manuscript authored by Sohn et al. reported the efficacy of a recombinant vaccine produced from plant N. benthamiana, against porcine epidemic diarrhea virus. Overall, the manuscript is well-written and organized with details making it easy to follow the authors’ claims and conclusions. However, there are some parts that need to be modified for improvement. Furthermore, it is believed that a high level of protection against PEDV is achieved by the first shot of a live-attenuated vaccine followed by a series of live-attenuated and/or killed vaccines in the field. In this regard, the authors need to discuss in the manuscript how this plant-derived vaccine could be used in this kind of vaccine program. Please refer to the comments below.

-Title: Although the manuscript includes the development of the plant-derived vaccine, it seems to focus on its protective efficacy when it was administrated into sows and then challenged their piglets with PEDV. Please consider changing the title.

-line 32: Replace ‘we’ to ‘We’.

-line 39: italicize ‘Alphacoronavirus.

-lines 64-67: Repetition from line 24-26. Consider rephrasing.

-section 2.1.: The plasmid includes any antibiotic resistance gene? Why the authors put 6x his tag? Any specific reason?

-line 76: Add the full name of N. benthamiana.

-line 77: Provide the type of PEDV and the homology of the encoding S1 protein sequence with the corresponding sequence of PEDV challenged in this study.

-line 97: Replace ‘Protein’ to ‘Recombinant protein’.

-section 2.5.: It seems redundant. The authors do not need to describe in detail how it was finalized unless the result is included.

-line 143: The ten non-challenged piglets used in any analysis?

-line 149: Change dpi to dpc.

-lines 159-160: Provide a PEDV-specific detection system.

-Table 1: add “-“ in score 4 on hyperemia in lamina propria.

-lines 198-199: It seems clinical sign scores and viral RNA quantification were not used in statistical analysis.

-lines 227-247: Modify the figure legend without line breaks.

-figure 1A: Add his-tag in the schematic drawing.

-figure 1D: Better to remove as the yield of rprotein was described in the manuscript.

-figure 2 and 3: Consider combining Fig. 2 and 3.

-section 3.2.and 3.3.: Add (Fig. 2) and (Fig. 3) in the manuscript.

-lines 273-274. Use a unified term for VNT or NA.

-figure 4 and 5: Consider combining them.

-line 285: Change viral DNA to viral RNA

-Table 2 and 3: Switch the numbers of figures.

-Table 2 and figure 6: Provide n numbers, how many samples were analyzed?

-line 320: The authors claimed “enhanced” immunogenicity of plant-derived recombinant vaccine used in this study, but it has been compared between vaccinated and non-vaccinated animals in this study. Thus, it could say the vaccine could elicit immunogenicity but not clear it is “enhanced”. Compared to what? Found the same description in lines 332-334. Please rephrase the expressions.

Lines 327-330: It is understandable that the rprotein used in this study is highly glycosylated based on the result from glycosidase treatment, but compared to the mammalian expression system, it is unclear which one would induce glycosylation more. The authors cited ref #6 in lines 219-220, and it said the S1 protein expressed from mammalian cells was 130 kDa, comparable to the rprotein expressed in the plant.

-line 358: Chonnam National University? No affiliation for that. Please correct it and the corresponding IACUC or study number approved.

Author Response

We are pleased to resubmit a revised manuscript (no. vaccines-2363436) entitled “Development of a plant-derived maternal vaccine against porcine epidemic diarrhea in piglets” for reconsideration in Journal of Vaccines published by MDPI as an original manuscript. We have carefully evaluated the reviewer’s comments and have provided a point-by-point response below. Changes in the manuscript have been identified by colored font. We hope that the revised manuscript meets the reviewers’ expectations at Journal of Vaccines.

Reviewer 2 Report

The manuscript submitted to Vaccines Manuscript ID: vaccines-2363436, entitled “Development of a plant-derived maternal vaccine against porcine epidemic diarrhea in piglets” is an interesting paper that used a novel vaccine approach by producing recombinant protein antigen in plants that was used to vaccinate pigs (sows) against PEDV. This vaccine was further modified to contain a FC receptor sequence for specific targeting to antigen presenting cells. Immunisation with this prototype vaccine induced lactogenic immunity with an increase in survival and general welfare of the piglets exposed to PEDV.  Plant based vaccines offer alternative immunisation strategies such as oral vaccination, an attractive model for future vaccine regimens.

General comments:

1.       It has been reported that the glycosylation of vaccine targets may impede the antigenicity and immunogenicity of the recombinant protein. In this study, it has been shown that glycosylation of the vaccine target occurs. Is it possible that the glycosylation may have contributed to the positive rates of virus of 63% at day 1 (and other time points of up to 100%) and the reduced mortality of 50%? If there were no glycosylation would you expect higher affinity antibodies to develop in the host that will also decrease the virus positive rates and increase the survival of the piglets? This should at least be addressed in the discussion.  

2.       No clear statement was made on if this animal trial was approved by an animal ethics committee and this statement should be included in the methods. Although mention was made of “institutional ethics in accordance of the Declaration of Helsinki”. However, the “Declaration of Helsinki” in my opinion is concerned with the ethical inclusion of human subjects in a study and not concerned with the welfare of the animals used in experiments (according to the Replacement, Reduction and Refinement principles). The numbers of animals used in the study also need a statistical validation method to be included in the methods, that indicate that the absolute minimum number of animals were used to achieve statistical significance (Reduction). Otherwise the numbers used may be deemed excessive from an ethical point of view.

Listed below are a few editorial suggestions for its improvement.

Combine the Figure one legend in one paragraph (Lines 225-236) so that there is a clear distinction between the text paragraph that starts at line 237 and the figure legend.

Figure 2 has three sub figures that is not numbered with A, B or C as was done for all the other composite figures. It also seem as if lines 251 – 253 should be part of figure legend and not a separate paragraph. (As wat was done for Figures 4 & 5)

Figure 3:  The legend is also split in two paragraphs and should be combined in one.

Figure 4: Please indicate if the data presented is representative of one animal or is this the average of the group? Also indicate the standard deviation of the NV-SOW data in figure 4A similar to what was done in Figure 5.

Author Response

(The authors gave the same response as above.)

Round 2

Reviewer 2 Report

The revisions provided by the authors are acceptable and improved the manuscript.